# Derivation and validation of a clinical severity score for acutely ill adults with suspected COVID-19: The PRIEST observational cohort study

Steve Goodacre[1]*, Ben Thomas[1], Laura Sutton[1], Matthew Burnsall[1], Ellen Lee[1], Mike Bradburn[1], Amanda Loban[1], Simon Waterhouse[1], Richard Simmonds[1], Katie Biggs[1], Carl Marincowitz[1], Jose Schutter[1], Sarah Connelly[1], Elena Sheldon[1], Jamie Hall[1], Emma Young[1], Andrew Bentley[2], Kirsty Challen[3], Chris Fitzsimmons[4], Tim Harris[5], Fiona Lecky[1], Andrew Lee[1], Ian Maconochie[6], Darren Walter[7]

1 School of Health and Related Research (ScHARR), University of Sheffield, Sheffield, United Kingdom,
2 Intensive Care, Manchester University NHS Foundation Trust, Wythenshawe Hospital, Manchester, United Kingdom, 3 Emergency Department, Lancashire Teaching Hospitals NHS Foundation Trust, Preston, United Kingdom, 4 Emergency Department, Sheffield Children's NHS Foundation Trust, Sheffield, United Kingdom, 5 Emergency Department, Barts Health NHS Trust, London, United Kingdom, 6 Emergency Department, Imperial College Healthcare NHS Trust, London, United Kingdom, 7 Emergency Department, Manchester University NHS Foundation Trust, Wythenshawe Hospital, Manchester, United Kingdom

* s.goodacre@sheffield.ac.uk

**Data Availability Statement:** Data are available in the in the ORDA data repository: http://doi.org/10.15131/shef.data.13194845.

## Abstract

### Objectives

We aimed to derive and validate a triage tool, based on clinical assessment alone, for predicting adverse outcome in acutely ill adults with suspected COVID-19 infection.

### Methods

We undertook a mixed prospective and retrospective observational cohort study in 70 emergency departments across the United Kingdom (UK). We collected presenting data from 22445 people attending with suspected COVID-19 between 26 March 2020 and 28 May 2020. The primary outcome was death or organ support (respiratory, cardiovascular, or renal) by record review at 30 days. We split the cohort into derivation and validation sets, developed a clinical score based on the coefficients from multivariable analysis using the derivation set, and the estimated discriminant performance using the validation set.

### Results

We analysed 11773 derivation and 9118 validation cases. Multivariable analysis identified that age, sex, respiratory rate, systolic blood pressure, oxygen saturation/inspired oxygen ratio, performance status, consciousness, history of renal impairment, and respiratory distress were retained in analyses restricted to the ten or fewer predictors. We used findings from multivariable analysis and clinical judgement to develop a score based on the NEWS2 score, age, sex, and performance status. This had a c-statistic of 0.80 (95% confidence

**Funding:** The PRIEST study was funded by the United Kingdom National Institute for Health Research Health Technology Assessment (HTA) programme (project reference 11/46/07). The funder played no role in the study design; in the collection, analysis, and interpretation of data; in the writing of the report; and in the decision to submit the article for publication. The views expressed are those of the authors and not necessarily those of the NHS, the NIHR or the Department of Health and Social Care.

**Competing interests:** All authors have completed the ICMJE uniform disclosure form at www.icmje. org/coi_disclosure.pdf and declare: grant funding to their employing institutions from the National Institute for Health Research; no financial relationships with any organisations that might have an interest in the submitted work in the previous three years; no other relationships or activities that could appear to have influenced the submitted work. This does not alter our adherence to PLOS ONE policies on sharing data and materials.

interval 0.79–0.81) in the validation cohort and predicted adverse outcome with sensitivity 0.98 (0.97–0.98) and specificity 0.34 (0.34–0.35) for scores above four points.

## Conclusion

A clinical score based on NEWS2, age, sex, and performance status predicts adverse outcome with good discrimination in adults with suspected COVID-19 and can be used to support decision-making in emergency care.

## Registration

ISRCTN registry, ISRCTN28342533, http://www.isrctn.com/ISRCTN28342533

## Introduction

The initial management of acutely ill people with suspected COVID-19 involves assessing the risk of adverse outcome and the need for life-saving intervention, to then determine decisions around hospital admission and inpatient referral [1–5]. Triage tools can assist decision-making by combining information from clinical assessment in a structured manner to predict the risk of adverse outcome. They can take the form of a score that increases with the predicted risk of adverse outcome or a rule that categorises patients into groups according to their risk or their intended management. Inclusion of laboratory and radiological information can improve prediction but requires hospital attendance, increases emergency department (ED) length of stay, and increases the infection risk related to repeated patient contacts. Triage tools also need to be applied prospectively to the relevant patient group, using the information available at the time of presentation. The limited availability of rapid tests with sufficient sensitivity to rule out COVID-19 at initial assessment means that the relevant population is suspected rather than confirmed COVID-19. An appropriate triage tool for COVID-19 therefore needs to be based on clinical assessment alone and applicable to people with suspected COVID-19.

We designed the Pandemic Influenza Triage in the Emergency Department (PAINTED) study following the 2009 H1N1 influenza pandemic to develop and evaluate triage tools in any future influenza pandemic [6]. We changed PAINTED to the Pandemic Respiratory Infection Emergency System Triage (PRIEST) study in January 2020 to address any pandemic respiratory infection, including COVID-19. The United Kingdom (UK) Department of Health and Social Care activated PRIEST on 20 March 2020 to develop and evaluate triage tools in the COVID-19 pandemic. Initial descriptive analysis of the PRIEST data showed that adults presenting to the ED with suspected COVID-19 have much higher rates of COVID-19 positivity, hospital admission and adverse outcome than children [7]. We therefore decided to undertake separate studies in adults and children, and only develop a new triage tools in adults, which we present here.

Evaluation of existing triage tools using the PRIEST study data suggested that CURB-65 [8], the National Early Warning Score version 2 (NEWS2) [9] and the Pandemic Modified Early Warning Score (PMEWS) [10] provide reasonable prediction for adverse outcome in suspected COVID-19 (c-statistics 0.75 to 0.77) [11]. Scope therefore existed to develop a specific triage tool for COVID-19 with better prediction for adverse outcome.

We aimed to derive and validate a triage tool in the form of an illness severity score, based on clinical assessment alone, for predicting adverse outcome in acutely ill adults with suspected COVID-19 infection.

## Materials and methods

We designed PRIEST as an observational study to collect standardised predictor variables recorded in the ED, which we would then use to derive and validate new tools for predicting adverse outcome up to 30 days after initial hospital presentation. The study did not involve any change to patient care. Hospital admission and discharge decisions were made according to usual practice, informed by local and national guidance.

We identified consecutive patients presenting to the ED of participating hospitals with suspected COVID-19 infection. Patients were eligible if they met the clinical diagnostic criteria [12] of fever (≥37.8˚C) and acute onset of persistent cough (with or without sputum), hoarseness, nasal discharge or congestion, shortness of breath, sore throat, wheezing, or sneezing. This was determined on the basis of the assessing clinician recording that the patient had suspected COVID-19 or completing a standardised assessment form designed for suspected pandemic respiratory infection [6]. During the study period COVID-19 testing was only recommended for those admitted to hospital, so it was recorded as a descriptive variable but not used to select patients or in the analysis.

For this study we planned to develop a triage tool in the form of an illness severity score based on clinical assessment and routine observations that any health care professional could use to rapidly estimate the risk of adverse outcome. The score would be based on a number of categorised variables, with points allocated to each category of each variable, which would then be summed to give a total score reflecting the predicted risk of adverse outcome. To enhance usability, we planned to (a) use a restricted number of variables, rather than all potentially predictive variables, and (b) categorise variables in accordance with currently used scores, unless there was clear evidence that these categories provided suboptimal prediction.

Data collection was both prospective and retrospective. Participating EDs were provided with a standardised data collection form (S1 Appendix) that included variables used in existing triage tools or considered to be potentially useful predictors of adverse outcome. Participating sites could adapt the form to their local circumstances, including integrating it into electronic or paper clinical records to facilitate prospective data collection, or using it as a template for research staff to retrospectively extract data from clinical records. We did not seek consent to collect data but information about the study was provided in the ED and patients could withdraw their data at their request. Patients with multiple presentations to hospital were only included once, using data from the first presentation identified by research staff.

Research staff at participating hospitals reviewed patient records at 30 days after initial attendance and recorded outcomes using the follow-up form in S2 Appendix. The primary outcome was death or major organ support (respiratory, cardiovascular, or renal) up to 30 days after initial attendance. Death and major organ support were also analysed separately as secondary outcomes. Our primary outcome definition reflected the need for triage tools to identify patients at risk of adverse outcome or requiring life-saving intervention to prevent adverse outcome. Respiratory support was defined as any intervention to protect the patient's airway or assist their ventilation, including non-invasive ventilation or acute administration of continuous positive airway pressure. It did not include supplemental oxygen alone or nebulised bronchodilators. Cardiovascular support was defined as any intervention to maintain organ perfusion, such as inotropic drugs, or invasively monitor cardiovascular status, such as central venous pressure or pulmonary artery pressure monitoring, or arterial blood pressure monitoring. It did not include peripheral intravenous cannulation or fluid administration. Renal support was defined as any intervention to assist renal function,

such as haemofiltration, haemodialysis, or peritoneal dialysis. It did not include intravenous fluid administration.

We randomly split the study population into derivation and validation cohorts by randomly allocating the participating sites to one or other cohort. We developed a score based on the prognostic value of predictor variables in multivariable analysis of the derivation cohort and expert judgements regarding clinical usability. Candidate predictors were combined in a multivariable regression with Least Absolute Shrinkage and Selection Operator (LASSO) using ten sample cross validation to select the model. The LASSO begins with a full model of candidate predictors and simultaneously performs predictor selection and penalisation during model development to avoid overfitting. The LASSO was performed twice: once where the number of predictors were unrestricted, and a second time when the LASSO was restricted to pick ten predictors. Fractional polynomials were used to model non-linear relationships for continuous variables.

We excluded cases from all analyses if age or outcome data were missing. We undertook three multivariable analyses, using different approaches to missing predictor variable data in the derivation cohort: (1) Complete case; (2) Multiple imputation using chained equations; (3) Deterministic imputation with missing predictor data assumed to be normal, where applicable. We did not consider any predictor with more than 50% missing data across the cohort for inclusion in the predictive model.

Clinical members of the research team reviewed the models and selected variables for inclusion in the triage tool, based on their prognostic value in the model, the clinical credibility of their association with adverse outcome, and their availability in routine clinical care. We categorised continuous variables, using recognised categories from existing scores where appropriate, while checking that categorisation reflected the relationship between the variable and adverse outcome in the derivation data. We then assigned integer values to each category of predictor variable, taking into account the points allocated to the category in existing scores, and the coefficient derived from a multivariable logistic regression model using categorised continuous predictors. This generated a composite clinical score in which risk of adverse outcome increased with the total score.

We applied the clinical score to the validation cohort, calculating diagnostic parameters at each threshold of the score, constructing a receiver-operating characteristic (ROC) curve, calculating the area under the ROC curve (c-statistic) and calculating the proportion with an adverse outcome at each level of the score. We used deterministic imputation to handle missing data in the validation cohort, assuming missing predictor variable data were normal, but excluding cases with more than a pre-specified number of predictor variables missing. We also undertook a complete case sensitivity analysis.

The sample size was dependent on the size and severity of the pandemic, but based on a previous study in the 2009 H1N1 influenza pandemic we estimated we would need to collect data from 20,000 patients across 40–50 hospitals to identify 200 (1%) with an adverse outcome, giving sufficient power for model derivation. In the event, the adverse outcome rate in adults was much higher in the COVID-19 pandemic (22%) [7], giving us adequate power to undertake derivation and validation of triage tools to predict all three outcomes.

## Patient and public involvement

The Sheffield Emergency Care Forum (SECF) is a public representative group interested in emergency care research [13]. Members of SECF advised on the development of the PRIEST study and two members joined the Study Steering Committee. Patients were not involved in

the recruitment to and conduct of the study. We are unable to disseminate the findings to study participants directly.

### Ethical approval

The North West—Haydock Research Ethics Committee gave a favourable opinion on the PAINTED study on 25 June 2012 (reference 12/NW/0303) and on the updated PRIEST study on 23rd March 2020, including the analysis presented here. The Confidentiality Advisory Group of the Health Research Authority granted approval to collect data without patient consent in line with Section 251 of the National Health Service Act 2006.

## Results

The PRIEST study recruited 22485 patients from 70 EDs across 53 sites between 26 March 2020 and 28 May 2020. We included 20889 in the analysis after excluding 39 who requested withdrawal of their data, 1530 children, 20 with missing outcome data, and seven with missing age. The derivation cohort included 11773 patients and the validation cohort 9118. Table 1 shows the characteristics of the derivation and validation cohorts. Around 31% of each cohort had COVID-19 confirmed, reflecting a combination of lack of testing in those discharged, suboptimal sensitivity of standard tests, and the difficulty of differentiating COVID-19 from similar presentations.

Table 2 shows summary statistics for each predictor variable in those with and without adverse outcome in the derivation sample, and univariate odds ratios for prediction of adverse outcome. Physiological variables were categorised to reflect their expected relationships with adverse outcome.

S1–S3 Tables show the results of multivariable analysis using complete case analysis, multiple imputation and deterministic imputation. Unrestricted LASSO on multiply imputed data included more predictors, with a higher c-statistic for the model (0.85, 95% CI 0.84 to 0.86), than the LASSO on deterministically imputed data or complete cases (c-statistics both 0.83, 95% CI 0.82 to 0.84). When restricted, there were nine predictors that were retained by LASSO in all three analyses (age, sex, respiratory rate, systolic BP, oxygen saturation/inspired oxygen ratio, history of renal impairment, performance status, consciousness and respiratory distress). C-statistics for the restricted models using deterministic imputation and complete case analysis (0.82, 95% CI 0.81 to 0.83) were slightly lower than c-statistics for the respective unrestricted models.

We developed a score through the following steps:

1. Clinical review judged that the nine predictors are clinically credible; that age, sex, respiratory rate, systolic BP, consciousness, oxygen saturation and inspired oxygen are routinely recorded in administrative systems and early warning scores (although the ratio of oxygen saturation to inspired oxygen is not routinely recorded); and that many EDs routinely record a measure of performance status for suspected COVID-19 cases that could be mapped onto our scale.

2. We decided to include temperature and heart rate, as these are routinely recorded alongside other physiological variables in early warning scores, and added prognostic value in the full models.

3. We created categories for age based on the observed multivariate association between age and outcome in our data, and categories for respiratory rate, heart rate, oxygen saturation, inspired oxygen, systolic BP, consciousness and temperature based on those used in the NEWS2 early warning score. NEWS2 is outlined in S3 Appendix.

**Table 1. Characteristics of the study population (derivation and validation cohorts).**

| Characteristic | Statistic/level | Derivation | Validation |
|---|---|---|---|
| Age (years) | N | 11773 | 9118 |
| | Mean (SD) | 62.4 (19.9) | 62.4 (19.5) |
| | Median (IQR) | 64 (48,79) | 64 (48,79) |
| Sex | Missing | 137 | 56 |
| | Male | 5746 (49.4%) | 4455 (49.2%) |
| | Female | 5890 (50.6%) | 4607 (50.8%) |
| Ethnicity | Missing/prefer not to say | 1819 | 2379 |
| | UK/Irish/other white | 8376 (84.1%) | 5867 (87.1%) |
| | Asian | 699 (7%) | 345 (5.1%) |
| | Black/African/Caribbean | 368 (3.7%) | 272 (4%) |
| | Mixed/multiple ethnic groups | 178 (1.8%) | 69 (1%) |
| | Other | 333 (3.3%) | 186 (2.8%) |
| Presenting features | Cough | 7248 (61.6%) | 5737 (62.9%) |
| | Shortness of breath | 8570 (72.8%) | 7000 (76.8%) |
| | Fever | 5714 (48.5%) | 4562 (50%) |
| Comorbidities | Hypertension | 3627 (30.8%) | 2807 (30.8%) |
| | Heart Disease | 2512 (21.3%) | 2188 (24%) |
| | Diabetes | 2394 (20.3%) | 1735 (19%) |
| | Asthma | 1867 (15.9%) | 1541 (16.9%) |
| | Other chronic lung disease | 2047 (17.4%) | 1717 (18.8%) |
| | Renal impairment | 1074 (9.1%) | 856 (9.4%) |
| | Active malignancy | 577 (4.9%) | 543 (6%) |
| | Immunosuppression | 312 (2.7%) | 319 (3.5%) |
| | Steroid therapy | 303 (2.6%) | 254 (2.8%) |
| | No chronic disease | 3385 (28.8%) | 2406 (26.4%) |
| Symptom duration (days) | N | 10790 | 8087 |
| | Mean (SD) | 8.1 (9.1) | 7.6 (8.6) |
| | Median (IQR) | 5 (2,10) | 5 (2,10) |
| Heart rate (beats/min) | N | 11506 | 8954 |
| | Mean (SD) | 94.7 (21.5) | 95.2 (21.7) |
| | Median (IQR) | 93 (80,108) | 94 (80,109) |
| Respiratory rate (breaths/min) | N | 11438 | 8908 |
| | Mean (SD) | 23.1 (6.9) | 23.4 (7.1) |
| | Median (IQR) | 22 (18,26) | 22 (18,26) |
| Systolic BP (mmHg) | N | 11423 | 8875 |
| | Mean (SD) | 134.5 (24.9) | 134.8 (25) |
| | Median (IQR) | 133 (118,149) | 133 (118,150) |
| Diastolic BP (mmHg) | N | 11373 | 8839 |
| | Mean (SD) | 78.3 (15.8) | 78.2 (16.5) |
| | Median (IQR) | 78 (68,88) | 78 (68,88) |
| Temperature (˚C) | N | 11307 | 8924 |
| | Mean (SD) | 37.1 (1.1) | 37.2 (1.1) |
| | Median (IQR) | 37 (36.4,37.8) | 37 (36.5,37.9) |
| Oxygen saturation (%) | N | 11658 | 8974 |
| | Mean (SD) | 94.9 (6.2) | 94.4 (7.5) |
| | Median (IQR) | 96 (94,98) | 96 (94,98) |
| Air or supplementary oxygen | Missing | 4113 | 4735 |

*(Continued)*

**Table 1.** (Continued)

| Characteristic | Statistic/level | Derivation | Validation |
|---|---|---|---|
| | On air | 5243 (68.4%) | 2544 (58%) |
| | On supplementary oxygen | 2417 (31.6%) | 1839 (42%) |
| Supplementary inspired oxygen (%) | N | 2417 | 1839 |
| | Mean (SD) | 45.9 (21.9) | 48.6 (22.5) |
| | Median (IQR) | 36 (28,60) | 36 (28,80) |
| Glasgow Coma Scale | N | 8627 | 6801 |
| | Mean (SD) | 14.6 (1.4) | 14.6 (1.4) |
| | Median (IQR) | 15 (15,15) | 15 (15,15) |
| Consciousness | Missing | 1515 | 872 |
| | Alert | 9774 (95.3%) | 7794 (94.5%) |
| | Verbal | 333 (3.2%) | 307 (3.7%) |
| | Pain | 101 (1%) | 82 (1%) |
| | Unresponsive | 50 (0.5%) | 63 (0.8%) |
| Performance status | Missing | 620 | 458 |
| | 1. Unrestricted normal activity | 5989 (53.7%) | 4547 (52.5%) |
| | 2. Limited strenuous activity, can do light activity | 1315 (11.8%) | 1056 (12.2%) |
| | 3. Limited activity, can self-care | 1565 (14%) | 1211 (14%) |
| | 4. Limited self-care | 1494 (13.4%) | 1155 (13.3%) |
| | 5. Bed/chair bound, no self-care | 790 (7.1%) | 691 (8%) |
| Admitted at initial assessment | Missing | 7 | 21 |
| | No | 3744 (31.8%) | 3122 (34.3%) |
| | Yes | 8022 (68.2%) | 5975 (65.7%) |
| Location of first admission† | Missing | 173 | 159 |
| | Ward | 7238 (92.2%) | 5409 (93%) |
| | ITU | 479 (6.1%) | 311 (5.3%) |
| | HDU | 132 (1.7%) | 96 (1.7%) |
| Respiratory pathogen | COVID-19 | 3660 (31.1%) | 2861 (31.4%) |
| | Influenza | 2 (0%) | 25 (0.3%) |
| | Other | 912 (7.7%) | 809 (8.9%) |
| | None identified | 7199 (61.1%) | 5423 (59.5%) |
| Mortality status | Missing | 0 | 3 |
| | Alive | 10002 (85%) | 7640 (83.8%) |
| | Dead | 1771 (15%) | 1475 (16.2%) |
| | Death with organ support* | 326 (18.4%) | 367 (24.9%) |
| | Death with no organ support* | 1445 (81.6%) | 1108 (75.1%) |
| Organ support | Respiratory | 939 (8%) | 1005 (11%) |
| | Cardiovascular | 316 (2.7%) | 201 (2.2%) |
| | Renal | 104 (0.9%) | 114 (1.3%) |
| | Any | 999 (8.5%) | 1059 (11.6%) |

* Denominator total deaths in category

† Denominator admitted patients

4. We created a multivariable logistic regression model using categorised predictor variables (S4 Table) and compared the coefficients for each category of predictor variable in the NEWS2 score to the points allocated in the NEWS2 score. We judged that the inconsistencies between the coefficients and the points used in NEWS2 were insufficient to justify

**Table 2. Univariate analysis of predictor variables for adverse outcome (derivation cohort).**

| Predictor Variable | Category (categorical variables) | n (outcome) | | Odds ratio | p-value | 95% CI |
|---|---|---|---|---|---|---|
| | | Adverse | Non adverse | | | |
| Age (n = 11773) | | | | 1.04 | 0.00 | (1.04, 1.04) |
| Sex (n = 11636) | Ref = Female | 1008 | 4882 | | | |
| | Male | 1413 | 4333 | 1.58 | 0.000 | (1.44, 1.73) |
| Ethnicity Category (n = 9954) | Ref = UK/Irish/other white | 1767 | 6609 | | | |
| | Asian | 138 | 561 | 0.92 | 0.399 | (0.76, 1.12) |
| | Black/African/Caribbean | 72 | 296 | 0.91 | 0.481 | (0.70, 1.18) |
| | Mixed/multiple ethnic groups | 28 | 150 | 0.70 | 0.084 | (0.46, 1.05) |
| | Other | 39 | 294 | 0.50 | 0.000 | (0.35, 0.70) |
| Shortness of breath (n = 11746) | Ref = No | 536 | 2640 | | | |
| | Yes | 1896 | 6674 | 1.40 | 0.000 | (1.26, 1.56) |
| Cough (n = 11746) | Ref = No | 1065 | 3433 | | | |
| | Yes | 1367 | 5881 | 0.75 | 0.000 | (0.68, 0.82) |
| Fever (n = 11746) | Ref = No | 1274 | 4758 | | | |
| | Yes | 1158 | 4556 | 0.95 | 0.253 | (0.87, 1.04) |
| Hypertension (n = 11732) | Ref = No | 1445 | 6660 | | | |
| | Yes | 995 | 2632 | 1.74 | 0.000 | (1.59, 1.91) |
| Heart Disease (n = 11732) | Ref = No | 1680 | 7540 | | | |
| | Yes | 760 | 1752 | 1.95 | 0.000 | (1.76, 2.15) |
| Diabetes (n = 11732) | Ref = No | 1733 | 7605 | | | |
| | Yes | 707 | 1687 | 1.84 | 0.000 | (1.66, 2.04) |
| Asthma (n = 11732) | Ref = No | 2143 | 7722 | | | |
| | Yes | 297 | 1570 | 0.68 | 0.000 | (0.60, 0.78) |
| Other chronic lung disease (n = 11732) | Ref = No | 1919 | 7766 | | | |
| | Yes | 521 | 1526 | 1.38 | 0.000 | (1.24, 1.54) |
| Renal impairment (n = 11732) | Ref = No | 2051 | 8607 | | | |
| | Yes | 389 | 685 | 2.38 | 0.000 | (2.09, 2.72) |
| Active malignancy (n = 11732) | Ref = No | 2248 | 8907 | | | |
| | Yes | 192 | 385 | 1.98 | 0.000 | (1.65, 2.36) |
| Immunosuppression (n = 11732) | Ref = No | 2360 | 9060 | | | |
| | Yes | 80 | 232 | 1.32 | 0.033 | (1.02, 1.71) |
| Steroid therapy (n = 11732) | Ref = No | 2363 | 9066 | | | |
| | Yes | 77 | 226 | 1.31 | 0.046 | (1.01, 1.70) |
| Symptom duration (n = 10790) | | | | 0.97 | 0.000 | (0.96, 0.98) |
| Number current medications (n = 11183) | | | | 1.09 | 0.00 | (1.08, 1.10) |
| Respiratory rate (n = 11773) | Ref = 12–20 or missing | 644 | 5061 | | | |
| | <9 | 3 | 5 | 4.72 | 0.034 | (1.12, 19.78) |
| | 9–11 | 3 | 8 | 2.95 | 0.111 | (0.78, 11.14) |
| | 21–24 | 581 | 2191 | 2.08 | 0.000 | (1.84, 2.36) |
| | >24 | 1213 | 2064 | 4.62 | 0.000 | (4.14, 5.15) |
| Systolic Blood Pressure (n = 11773) | Ref = 111–219 or missing | 1860 | 8093 | | | |
| | 101–110 | 269 | 745 | 1.57 | 0.000 | (1.35, 1.82) |
| | 91–100 | 170 | 320 | 2.31 | 0.000 | (1.91, 2.80) |
| | <91 | 137 | 143 | 4.17 | 0.000 | (3.28, 5.30) |
| | >219 | 8 | 28 | 1.24 | 0.588 | (0.57, 2.73) |
| Heart rate (n = 11773) | Ref = 51–90 or missing | 1007 | 4353 | | | |
| | <41 | 15 | 42 | 1.54 | 0.152 | (0.85, 2.79) |

*(Continued)*

**Table 2.** (Continued)

| Predictor Variable | Category (categorical variables) | n (outcome) | | Odds ratio | p-value | 95% CI |
|---|---|---|---|---|---|---|
| | | Adverse | Non adverse | | | |
| | 41–50 | 12 | 42 | 1.24 | 0.521 | (0.65, 2.35) |
| | 91–110 | 776 | 3112 | 1.08 | 0.159 | (0.97, 1.20) |
| | 111–130 | 450 | 1367 | 1.42 | 0.000 | (1.25, 1.62) |
| | >130 | 184 | 413 | 1.93 | 0.000 | (1.60, 2.32) |
| Temperature (n = 11773) | Ref = 36.1–38.0 or missing | 1498 | 6747 | | | |
| | 35.1–36 | 245 | 958 | 1.15 | 0.067 | (0.99, 1.34) |
| | 38.1–39 | 446 | 1137 | 1.77 | 0.000 | (1.56, 2.00) |
| | >39.0 | 166 | 386 | 1.94 | 0.000 | (1.60, 2.34) |
| | <35.1 | 89 | 101 | 3.97 | 0.000 | (2.97, 5.31) |
| GCS Total (n = 8627) | Ref = Mild (13–15) | 1551 | 6618 | | | |
| | Moderate (9–12) | 187 | 150 | 5.32 | 0.000 | (4.26, 6.64) |
| | Severe (< = 8) | 73 | 48 | 6.49 | 0.000 | (4.49, 9.38) |
| AVPU (n = 10258) | Ref = Alert | 1756 | 8018 | | | |
| | Verbal | 176 | 157 | 5.12 | 0.000 | (4.10, 6.39) |
| | Pain | 62 | 39 | 7.26 | 0.000 | (4.85, 10.87) |
| | Unresponsive | 32 | 18 | 8.12 | 0.000 | (4.55, 14.49) |
| Performance status (n = 11153) | Ref = Unrestricted normal activity | 709 | 5280 | | | |
| | Limited strenuous activity, can do light activity | 268 | 1047 | 1.91 | 0.000 | (1.63, 2.23) |
| | Limited activity, can self care | 430 | 1135 | 2.82 | 0.000 | (2.46, 3.23) |
| | Limited self care | 560 | 934 | 4.47 | 0.000 | (3.92, 5.09) |
| | Bed/chair bound, no self care | 334 | 456 | 5.45 | 0.000 | (4.64, 6.41) |
| Severe respiratory distress (n = 11773) | Ref = No | 2250 | 9184 | | | |
| | Yes | 194 | 145 | 5.46 | 0.000 | (4.38, 6.81) |
| Respiratory exhaustion (n = 11773) | Ref = No | 2360 | 9227 | | | |
| | Yes | 84 | 102 | 3.22 | 0.000 | (2.40, 4.31) |
| Severe dehydration (n = 11773) | Ref = No | 2373 | 9240 | | | |
| | Yes | 71 | 89 | 3.11 | 0.000 | (2.27, 4.26) |
| Previous attendance (n = 11773) | Ref = No | 2160 | 8429 | | | |
| | Yes | 284 | 900 | 1.23 | 0.004 | (1.07, 1.42) |
| Known contact with Covid-19 case (n = 1177 | Ref = No | 2175 | 8474 | | | |
| | Yes | 269 | 855 | 1.23 | 0.006 | (1.06, 1.42) |
| Central capillary refill (n = 2935) | Ref = Normal | 486 | 2179 | | | |
| | Abnormal | 101 | 169 | 2.68 | 0.000 | (2.05, 3.49) |

allocating alternative points in our score. We allocated points to categories of age, sex, performance status, renal history, and respiratory distress, based on the coefficients in the model.

5. We removed renal history and respiratory distress from the multivariable model (S5 Table), noted that this made no meaningful difference to the c-statistic (0.82 in both models) and, given concerns about subjectivity and lack of routine recording, decided not to include them in the score.

Fig 1 provides a summary of the derivation process. The developed score is shown in Fig 2. We applied the score to the validation cohort. Fig 3 shows the ROC curve, with a c-statistic of 0.80 (95% CI 0.79 to 0.81) for the score. Sensitivity analysis using only complete cases gave a c-statistic of 0.79 (95% CI 0.77 to 0.80). S1 and S2 Figs show the calibration plots for the

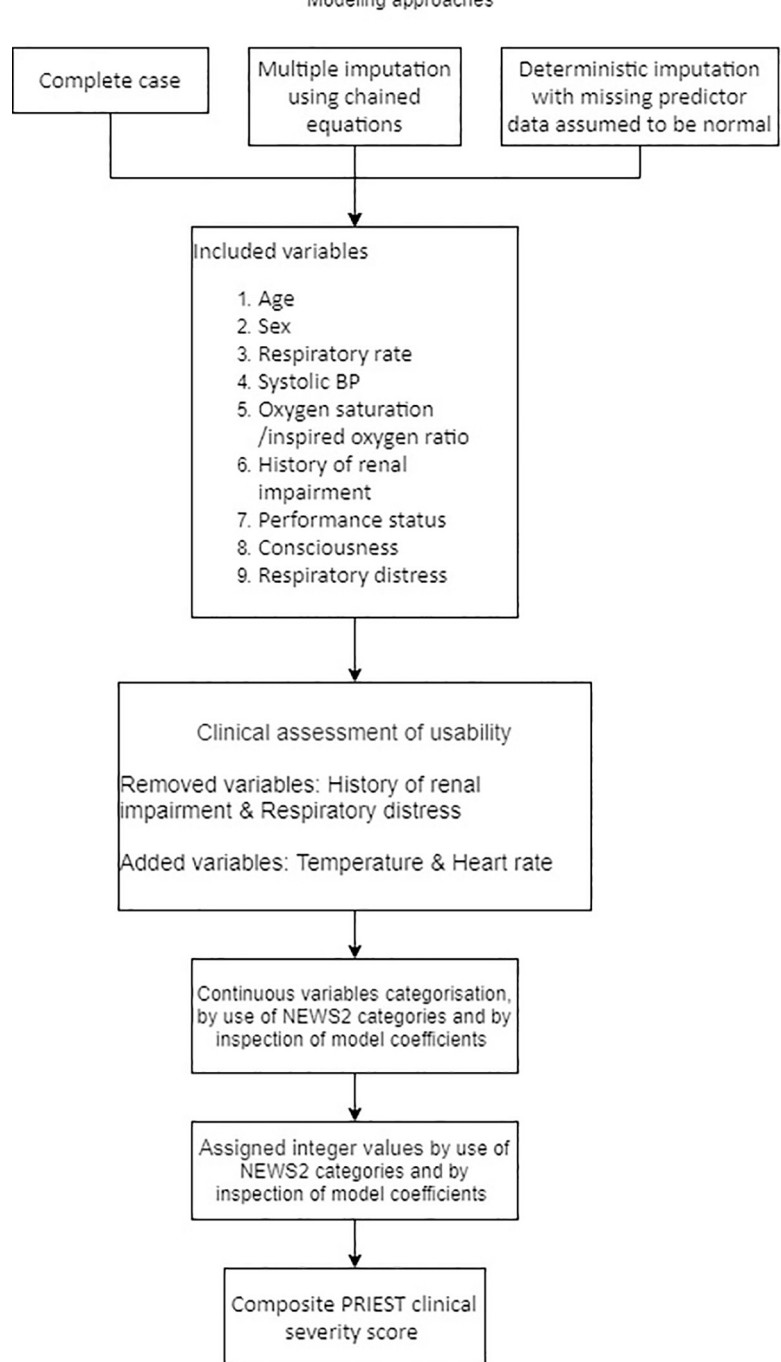

**Fig 1. Summary of the derivation process.**

unrestricted and restricted LASSO models applied to the validation cohort. The c-statistics (0.82 and 0.81 respectively, compared with 0.80 for the score) indicate the effect of restricting the number of variables and then developing a score had upon discrimination. Fig 4 shows the probability of adverse outcome for each value of the score. Table 3 shows the sensitivity and specificity for predicting outcome at each threshold of the triage tool.

| Variable | Range | Score |
|---|---|---|
| Respiratory rate (per minute) | 12-20 | 0 |
| | 9-11 | 1 |
| | 21-24 | 2 |
| | <9 or >24 | 3 |
| Oxygen saturation (%) | >95 | 0 |
| | 94-95 | 1 |
| | 92-93 | 2 |
| | <92 | 3 |
| Heart rate (per minute) | 51-90 | 0 |
| | 41-50 or 91-110 | 1 |
| | 111-130 | 2 |
| | <41 or >130 | 3 |
| Systolic BP (mmHg) | 111-219 | 0 |
| | 101-110 | 1 |
| | 91-100 | 2 |
| | <91 or >219 | 3 |
| Temperature (°C) | 36.1-38.0 | 0 |
| | 35.1-36.0 or 38.1-39.0 | 1 |
| | >39.0 | 2 |
| | <35.1 | 3 |
| Alertness | Alert | 0 |
| | Confused or not alert | 3 |
| Inspired oxygen | Air | 0 |
| | Supplemental oxygen | 2 |
| Sex | Female | 0 |
| | Male | 1 |
| Age (years) | 16-49 | 0 |
| | 50-65 | 2 |
| | 66-80 | 3 |
| | >80 | 4 |
| Performance status | Unrestricted normal activity | 0 |
| | Limited strenuous activity, can do light activity | 1 |
| | Limited activity, can self-care | 2 |
| | Limited self-care | 3 |
| | Bed/chair bound, no self-care | 4 |

**Fig 2. The PRIEST COVID-19 clinical severity score.**

S3 and S4 Figs show the ROC curves, and S6 and S7 Tables show the predictive performance of the score when applied to the secondary outcomes of organ support and death without organ support in the validation cohort. The score provided better prognostic discrimination for death without organ support (c-statistic 0.83, 95% CI 0.82 to 0.84) than for organ support (0.68, 95% CI 0.67 to 0.69).

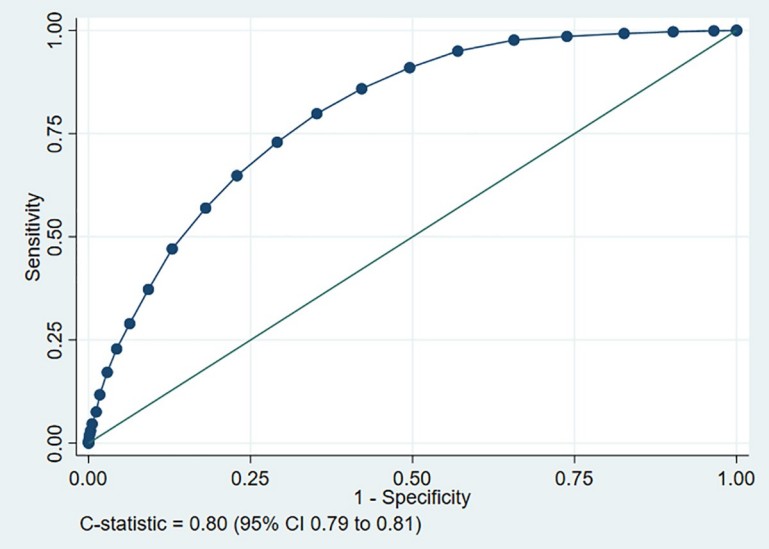

**Fig 3. ROC curve for the tool predicting the primary outcome of death or organ support, validation cohort.**

## Discussion

We have developed a clinical illness severity score for acutely ill patients presenting to the ED with suspected COVID-19 that combines the NEWS2 score, age, sex, and performance status to predict the risk of death or receipt of organ support in the following 30 days. The score ranges from zero to 29 points, with a score greater than four predicting adverse outcome with high sensitivity and low specificity. In developing the score, we tried to optimise usability without compromising performance. Usability was optimised by basing the score on the existing NEWS2 score and only adding easily available information. The c-statistic of the score on the validation cohort was 0.80, compared with 0.82 and 0.81 when the unrestricted and restricted

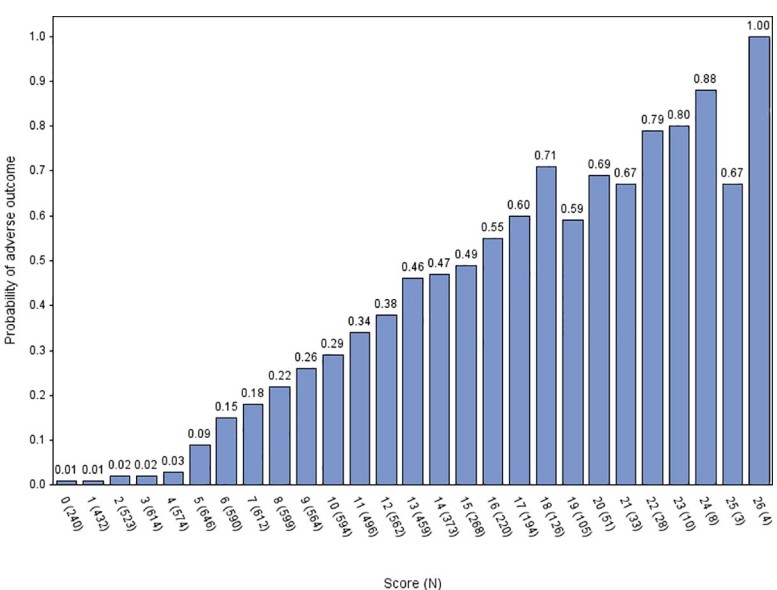

**Fig 4. Probability of adverse outcome for each value of the score, validation cohort.**

**Table 3. Sensitivity, specificity, PPV, NPV and proportion with a positive score at each score threshold for predicting the primary outcome of death or organ support, validation cohort.**

| Score threshold | Proportion with positive score | Sensitivity (95% CI) | Specificity (95% CI) | Positive predictive value (95% CI) | Negative predictive value (95% CI) |
|---|---|---|---|---|---|
| >0 | 0.97 | 1.00 (1.00, 1.00) | 0.04 (0.03,0.04) | 0.25 (0.24, 0.25) | 0.99 (0.98, 1.00) |
| >1 | 0.92 | 1.00 (1.00, 1.00) | 0.10 (0.10, 0.10) | 0.26 (0.25, 0.26) | 0.99 (0.99, 0.99) |
| >2 | 0.87 | 0.99 (0.99, 0.99) | 0.17 (0.17, 0.18) | 0.27 (0.27, 0.28) | 0.99 (0.98, 0.99) |
| >3 | 0.80 | 0.99 (0.98, 0.99) | 0.26 (0.26, 0.27) | 0.30 (0.29, 0.30) | 0.98 (0.98, 0.99) |
| >4 | 0.73 | 0.98 (0.97, 0.98) | 0.34 (0.34, 0.35) | 0.32 (0.31, 0.32) | 0.98 (0.98, 0.98) |
| >5 | 0.66 | 0.95 (0.95, 0.95) | 0.43 (0.43, 0.43) | 0.34 (0.34, 0.35) | 0.96 (0.96, 0.97) |
| >6 | 0.59 | 0.91 (0.91, 0.91) | 0.50 (0.50, 0.51) | 0.37 (0.36, 0.37) | 0.95 (0.94, 0.95) |
| >7 | 0.53 | 0.86 (0.85, 0.86) | 0.58 (0.57, 0.58) | 0.39 (0.39, 0.40) | 0.93 (0.93, 0.93) |
| >8 | 0.46 | 0.80 (0.79, 0.80) | 0.65 (0.64, 0.65) | 0.42 (0.41, 0.42) | 0.91 (0.91, 0.91) |
| >9 | 0.40 | 0.73 (0.72, 0.74) | 0.71 (0.71, 0.71) | 0.44 (0.44, 0.45) | 0.89 (0.89, 0.90) |
| >10 | 0.33 | 0.65 (0.64, 0.66) | 0.77 (0.77, 0.77) | 0.47 (0.46, 0.48) | 0.87 (0.87, 0.88) |
| >11 | 0.27 | 0.57 (0.56, 0.58) | 0.82 (0.82, 0.82) | 0.50 (0.49, 0.50) | 0.86 (0.86, 0.86) |
| >12 | 0.21 | 0.47 (0.46, 0.48) | 0.87 (0.87, 0.87) | 0.53 (0.53, 0.54) | 0.84 (0.84, 0.84) |
| >13 | 0.16 | 0.37 (0.37, 0.38) | 0.91 (0.91, 0.91) | 0.56 (0.55, 0.57) | 0.82 (0.82, 0.82) |
| >14 | 0.12 | 0.29 (0.28, 0.30) | 0.94 (0.93, 0.94) | 0.59 (0.58, 0.60) | 0.81 (0.80, 0.81) |
| >15 | 0.09 | 0.23 (0.22, 0.23) | 0.96 (0.95, 0.96) | 0.62 (0.61, 0.64) | 0.80 (0.79, 0.80) |
| >16 | 0.06 | 0.17 (0.17, 0.18) | 0.97 (0.97, 0.97) | 0.65 (0.64, 0.67) | 0.79 (0.79, 0.79) |
| >17 | 0.04 | 0.12 (0.11, 0.12) | 0.98 (0.98, 0.98) | 0.68 (0.66, 0.70) | 0.78 (0.78, 0.78) |
| >18 | 0.03 | 0.08 (0.07, 0.08) | 0.99 (0.99, 0.99) | 0.67 (0.65, 0.69) | 0.77 (0.77, 0.78) |
| >19 | 0.02 | 0.05 (0.04, 0.05) | 0.99 (0.99, 1.00) | 0.73 (0.70, 0.76) | 0.77 (0.77, 0.77) |
| >20 | 0.01 | 0.03 (0.03, 0.03) | 1.00 (1.00, 1.00) | 0.76 (0.72, 0.79) | 0.77 (0.76, 0.77) |
| >21 | 0.01 | 0.02 (0.02, 0.02) | 1.00 (1.00, 1.00) | 0.81 (0.76, 0.85) | 0.76 (0.76, 0.77) |
| >22 | 0.00 | 0.01 (0.01, 0.01) | 1.00 (1.00, 1.00) | 0.84 (0.76, 0.90) | 0.76 (0.76, 0.77) |
| >23 | 0.00 | 0.01 (0.00, 0.01) | 1.00 (1.00, 1.00) | 0.87 (0.76, 0.94) | 0.76 (0.76, 0.76) |
| >24 | 0.00 | 0.00 (0.00, 0.00) | 1.00 (1.00, 1.00) | 0.86 (0.66, 0.96) | 0.76 (0.76, 0.76) |

models were applied to the validation cohort, suggesting that simplifying the tool did not excessively compromise prediction.

Our score has a number of features that sets it apart from other scores. Derivation and validation were rigorously undertaken, following an independently peer-reviewed protocol set up in advance of the pandemic, using data from a very large and representative cohort presenting to EDs across the UK, and analysed using a pre-specified statistical analysis plan. Our choice of adverse outcome ensured that the score predicts need for life-saving intervention, not just mortality. Our patient selection criteria ensure that the score is applicable to the clinically relevant population of suspected COVID-19 rather than a confirmed cohort, which would typically be assembled retrospectively and exclude those with diagnostic uncertainty at presentation. We also included patients who were discharged after ED assessment, which is essential if the score is to be used to support decision-making around admission or discharge.

Our score improves upon those recommended in existing guidelines for the initial assessment of suspected acute COVID-19, with a c-statistic of 0.8 compared to 0.75 for CURB-65 and 0.77 for NEWS2 and PMEWS [11]. The practical implications of improved prediction can be appreciated by considering how the addition of age, sex and performance status to NEWS2 might improve decision-making around admission. NEWS2 would suggest that a young person with unlimited performance status and an elderly person with limited performance but the same NEWS2 score should have the same admission decision, whereas our score

recognises that safe discharge is much more likely to be achieved in the younger patient. Our score shares similarities with PMEWS, which was developed for the H1N1 influenza pandemic, but achieves better prediction by using more detailed age, sex and performance status data.

Since the start of the pandemic numerous studies have sought to develop and evaluate prediction scores for COVID-19. A living systematic review [14] has identified 50 prognostic models for adverse outcome in people with diagnosed COVID-19. C-statistics ranged from 0.68 to 0.99, and the most frequently used predictor variables were age, sex, comorbidities, temperature, lymphocyte count, C reactive protein, creatinine, and imaging features. Recently the ISARIC WHO Clinical Characterisation Protocol developed and validated the 4C Mortality Score [15] that predicts the mortality risk for people admitted with COVID-19 with better discriminant performance than 15 pre-existing risk stratification scores (c-statistic 0.77 versus 0.61–0.76).

These scores have important limitations as triage tools, which we have attempted to address in developing our triage tool. Many were developed to predict mortality, whereas triage tools need to predict need for life-saving treatment. Most were developed on admitted populations, whereas the relevant population for an initial assessment tool needs to include those discharged after assessment. This is because the decision to be admit is likely to be based upon the same predictor variables that are used in the tool, so excluding discharged patients will underestimate the predictive value of these variables. For example, oxygen saturation is an important predictor of adverse outcome and is also an important criterion for determining hospital admission. Developing a triage tool on a population selected on the basis of oxygen saturation will underestimate the value of oxygen saturation as a predictor. This may explain why many scores developed on admitted patients do not include well-recognised clinical predictors. Finally, inclusion of laboratory data as predictor variables prolongs ED stay and prevents the triage tool being used for rapid assessment.

Rapid clinical scores have been proposed or evaluated in several studies. Liao *et al* [16] proposed adding age>65 years to the NEWS2 score to aid decision-making, based on early experience of the pandemic in China. Myrstad *et al* [17] reported a c-statistic of 0.822 (95% CI 0.690 to 0.953) for NEWS2 predicting death or severe disease in a small study (N = 66) of people hospitalised with confirmed COVID-19. Hu *et al* [18] reported c-statistics of 0.833 (0.737 to 0.928) for the Rapid Emergency Medicine Score (REMS) and 0.677 (0.541 to 0.813) for the Modified Emergency Medicine Score (MEWS) for predicting mortality in critically ill patients with COVID-19. Haimovich *et al* [19] developed the quick COVID-19 severity index, consisting of respiratory rate, oxygen saturation, and oxygen flow rate, which predicted respiratory failure within 24 hours in adults admitted with COVID-19 requiring supplemental oxygen with a c-statistic of 0.81 (0.73 to 0.89). These studies are limited by small numbers (producing imprecise estimates of accuracy), single-centre design (limiting generalisability) and only including admitted patients.

An important limitation of our study is that retrospective data collection resulted in some missing and may have resulted in some inaccuracy of predictor variable recording. Recording of inspired oxygen concentration was subject to a particularly high rate of missing data. We anticipated this problem and pre-specified analyses involving multiple imputation, deterministic imputation, and complete case analysis to explore the impact of missing data. There was reasonable concordance between the models. Another potential limitation is that our definition of adverse outcome did not include events occurring after 30 days or requirements for hospital admission (such as oxygen therapy or intravenous fluids) that fell short of our definition of organ support. We may also have missed adverse outcomes if patients attended a different hospital after initial hospital discharge. This is arguably less likely in the context of a

pandemic, in which movements between regions were curtailed, but cannot be discounted. The 5-point scale we used for determining performance status has not been widely used or evaluated, although the 9-point clinical frailty index maps onto it reasonably well. Finally, although our triage tool can be used in the prehospital or community setting, we recommend caution in extrapolating our findings to settings where there is likely to be a lower prevalence of adverse outcome.

Our clinical score could be used to support ED decision-making around hospital admission and inpatient referral. Scores of four or less could identify a proportion of patients at low risk of adverse outcome who would be suitable for discharge home, while a higher threshold could be used to select patients for critical care. However, triage tools should only support and not replace clinical decision-making. The clinical context, patient preferences, and available resources must be considered. This may be illustrated by older patients (especially male) with limited performance status who score greater than four with little or no physiological abnormalities. These patients would not necessarily be at high risk of adverse outcome if they were managing their symptoms at home but the clinical context is presentation to a hospital ED. Our data show that if these patients needed ED assessment then they were at significant risk of adverse outcome even if there was little physiological abnormality. In terms of decision-making, patient preference should be taken into account, since these patients may accept discharge with a significant risk of adverse outcome if hospital admission provides no clear benefit.

Our triage tool could also be used to support prehospital and community decision-making around decisions to refer for hospital assessment. However, the importance of developing scores in an appropriate population needs to be considered. A score developed on an ED population may be inappropriate for supporting decisions to transport to the ED in the same way as scores developed on the inpatient population may be inappropriate for supporting admission decisions in an ED population. Further validation is required to determine the performance of the tool in these settings. Further ED validation in subsequent waves of the pandemic or other ED settings would also be helpful to determine whether changes or differences in the pandemic population or outcomes lead to changes in outcome prediction.

In summary, we have developed a clinical score that can provide a rapid and accurate assessment of the risk of adverse outcome in adults who are acutely ill with suspected COVID-19.

## Supporting information

**S1 Fig. Calibration plot for unrestricted LASSO model performance, validation cohort.**
(TIF)

**S2 Fig. Calibration plot for restricted LASSO model performance, validation cohort.**
(TIF)

**S3 Fig. ROC curve for the tool predicting the secondary outcome of organ support, validation cohort.**
(TIF)

**S4 Fig. ROC curve for tool predicting the secondary outcome of death without organ support, validation cohort.**
(TIF)

**S1 Table. Multivariable analysis, complete case (N = 5988).**
(DOCX)

**S2 Table. Multivariable analysis, using multiple imputation (50 imputations; N = 11636).**
(DOCX)

**S3 Table. Multivariable analysis, using deterministic imputation (N = 9891).**
(DOCX)

**S4 Table. Logistic regression model based on selected categorised predictor variables.**
(DOCX)

**S5 Table. Logistic regression model based on selected categorised predictor variables, excluding respiratory distress and history of renal impairment.**
(DOCX)

**S6 Table. Sensitivity, specificity, PPV, NPV at each score threshold for predicting the secondary outcome of organ support, validation cohort.**
(DOCX)

**S7 Table. Sensitivity, specificity, PPV, NPV at each score threshold for predicting the secondary outcome of death without organ support, validation cohort.**
(DOCX)

**S1 Appendix. Standardised data collection form.**
(PDF)

**S2 Appendix. Follow-up form.**
(PDF)

**S3 Appendix. The NEWS2 score.**
(DOCX)

**S4 Appendix. Study steering committee.**
(DOCX)

**S5 Appendix. Site research staff.**
(DOCX)

**S6 Appendix. Supporting research staff.**
(DOCX)

## Acknowledgments

We thank Katie Ridsdale for clerical assistance with the study, Erica Wallis (Sponsor representative, all members of the Study Steering Committee (S4 Appendix) and the site research teams who delivered the data for the study (S5 Appendix), and the research team at the University of Sheffield past and present (S6 Appendix).

## Author Contributions

**Conceptualization:** Steve Goodacre, Andrew Bentley, Kirsty Challen, Chris Fitzsimmons, Tim Harris, Fiona Lecky, Andrew Lee, Ian Maconochie, Darren Walter.

**Data curation:** Ben Thomas, Laura Sutton, Amanda Loban, Simon Waterhouse, Richard Simmonds, Jose Schutter, Sarah Connelly, Elena Sheldon, Jamie Hall, Emma Young.

**Formal analysis:** Steve Goodacre, Ben Thomas, Laura Sutton, Matthew Burnsall, Ellen Lee.

**Funding acquisition:** Steve Goodacre, Andrew Bentley, Kirsty Challen, Chris Fitzsimmons, Tim Harris, Fiona Lecky, Andrew Lee, Ian Maconochie, Darren Walter.

**Investigation:** Carl Marincowitz.

**Methodology:** Mike Bradburn.

**Project administration:** Ben Thomas, Katie Biggs.

**Validation:** Matthew Burnsall, Ellen Lee.

**Writing – original draft:** Steve Goodacre.

**Writing – review & editing:** Ben Thomas, Laura Sutton, Matthew Burnsall, Ellen Lee, Mike Bradburn, Amanda Loban, Simon Waterhouse, Richard Simmonds, Katie Biggs, Carl Marincowitz, Jose Schutter, Sarah Connelly, Elena Sheldon, Jamie Hall, Emma Young, Andrew Bentley, Kirsty Challen, Chris Fitzsimmons, Tim Harris, Fiona Lecky, Andrew Lee, Ian Maconochie, Darren Walter.

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
