## [Decision Letter · Decision Letter 0]

11 Dec 2020

PONE-D-20-34978

Derivation and validation of a clinical severity score for acutely ill adults with suspected COVID-19: The PRIEST observational cohort study

PLOS ONE

Dear Dr. Goodacre,

Thank you for submitting your manuscript to PLOS ONE. After careful consideration, we feel that it has merit but does not fully meet PLOS ONE’s publication criteria as it currently stands. Therefore, we invite you to submit a revised version of the manuscript that addresses the points raised during the review process.

I do wish to emphasize one point, the authors need to emphasize their uniqueness among various other scores recently published. 

We look forward to receiving your revised manuscript.

Kind regards,

Itamar Ashkenazi

Academic Editor

PLOS ONE

Journal Requirements:

2.Please amend the manuscript submission data (via Edit Submission) to include author Ben Thomas, Laura Sutton, Matthew Burnsall, Ellen Lee, Mike Bradburn, Amanda Loban, Simon Waterhouse, Richard Simmonds, Katie Biggs, Carl Marincowitz, Jose Schutter, Sarah Connelly, Elena Sheldon, Jamie Hall, Emma Young, Andrew Bentley, Kirsty Challen, Chris Fitzsimmons, Tim Harris, Fiona Lecky, Andrew Lee, Ian Maconochie, Darren

Walter.

3.Thank you for stating the following in the Competing Interests section:

[All authors have completed the ICMJE uniform disclosure form at www.icmje.org/coi_disclosure.pdf and declare: grant funding to their employing institutions from the National Institute for Health Research; no financial relationships with any organisations that might have an interest in the submitted work in the previous three years; no other relationships or activities that could appear to have influenced the submitted work.].

4.Thank you for submitting the above manuscript to PLOS ONE. During our internal evaluation of the manuscript, we found significant text overlap between your submission and the following previously published works:

- https://www.medrxiv.org/content/10.1101/2020.09.02.20185892v1

- http://eprints.whiterose.ac.uk/165084/1/2020.08.10.20171496v1.full.pdf

Please revise the manuscript and tables to rephrase or remove the duplicated text, cite your sources, and provide details as to how the current manuscript advances on previous work. Please note that further consideration is dependent on the submission of a manuscript that addresses these concerns about the overlap in text with published work.

Reviewers' comments:

Reviewer's Responses to Questions

**Comments to the Author**

1. Is the manuscript technically sound, and do the data support the conclusions?

Reviewer #1: Partly

Reviewer #2: Yes

2. Has the statistical analysis been performed appropriately and rigorously? 

Reviewer #1: Yes

Reviewer #2: Yes

3. Have the authors made all data underlying the findings in their manuscript fully available?

Reviewer #1: Yes

Reviewer #2: Yes

4. Is the manuscript presented in an intelligible fashion and written in standard English?

Reviewer #1: Yes

Reviewer #2: Yes

5. Review Comments to the Author

Reviewer #1: The authors have derived a severity of illness / probability of adverse outcome score for patients with suspected COVID-19, who presented to hospital. Their data was derived from 70 centres in the UK and included roughly 22,500 individual patient initial assessments. The primary outcome was death or ongoing organ support at 30 days. 15% of the patients had this outcome. Their score combines NEWS2, age , sex and performance status.

General comments

1. The patient data was derived during the first wave of COVID-19 in the UK. The second wave in the UK, and elsewhere, appears to have some different characteristics. Could the authors comment on this and / or speculate on the need to repeat their data collection in other countries / during the current (second) wave, to establish whether it c statistic is affected?

2. Might the authors data have a significant selection bias in that the location of data collection was emergency departments? Patients will have presented to primary care, Acute Medical Units etc etc hence how certain can we be that this result is generalisable?

3. Do the authors have any data to suggest what the short-term outcome of their patients was - e.g. how many were receiving level 1, level 2 and level 3 care at 72 hours post presentation? If not, might this secondary outcome be useful in future studies?

4. Could the authors please include the data for patient sex in Table 2?

5. Could the authors comment on the fact that ~67% of patients were admitted but only 31% were SARS-CoV-2 positive?

6. Could the authors comment further on the very high proportion of missing data related to supplemental oxygen, and the much larger proportion in the validation cohort?

7. Could the authors please provide more detail in figure 3 , in particular, the number of patients with each score and the probabilities for each score 17-29?

8. Given that the authors cut off for risk of a "bad outcome" is stated as a score >4 / probability of >9%, which can be achieved by being a 50 year old male with a temperature of 38.1 and a heart rate of 91 seems poorly calibrated to real life?

9. As currently written, I am unconvinced that the authors addition of age, sex and performance status adds anything useful to NEWS2 as a triage tool, especially in the absence of knowing whether the patient is positive for SARS-CoV-2. Perhaps they could clarify their justification for ading these variables to their real-world, pre-diagnosis target population?

Reviewer #2: In this study, the authors aimed to develop and validate a triage tool, based on clinical assessment alone, for predicting death or organ support at Day-30 in acutely ill adults with suspected COVID-19 infection. In the first part of the study (derivation cohort of 11773 patients), using multivariable analysis, the authors identified a restricted number of variables with the best prognostic value, clinical relevance and availability and then assigned integer values to each, resulting in a composite score in which the higher the value, the poorer the prognosis. In the second part of the study (validation cohort of 9118 patients), the authors tested their score and its ability to predict adverse outcomes. They found that a score including the NEWS2 score, age, sex and performance status could predict Day-30 adverse outcomes with high sensitivity but low specificity. The study is well-written and easy to read. Methods are well described and explained and the statistical analysis is appropriate. Results are interesting and the score could be useful in clinical practice in the event of a third pandemic wave. The two main strengths of the study are the large sample size and the appropriate method for building the score. One of the main limitations is the added-value of this new score, compared to existing triage scores for patients admitted to emergency department. I have some additional concerns that need to be discussed.

1. Why did you consider patients with suspected and not confirmed COVID-19? In fact, this score could be used for all patients admitted to the emergency department, regardless of their COVID-19 status and is therefore not very different from existing triage scores. It would have been more interesting to develop a specific score for COVID-19 patients. Please clarify this point. In this regard, it would be interesting to repeat the analysis only in patients with confirmed COVID-19 if you are convinced of the need to develop specific predictive scores for COVID-19 patients.

2. What was the proportion of patients in whom COVID-19 was confirmed? This is a very important point before discussing your results, since the rationale for your study is based on the need to develop scores for COVID-19. Your results cannot be considered in the same way according to the proportion of confirmed COVID-19 patients.

3. You state in the opening of the discussion that you included the NEWS2 score in your own score in addition to age, sex and performance status. However, you never described the NEWS2 score in the manuscript and in the Figure 1 (which is a Table), which is very confusing for readers. From my point of view, it would be clearer to simply indicate all the variable you included in your score rather than talking about the News2 score. Please clearly explain what the NEWS2 score is and clarify this point.

4. In figure 2, since your score ranged from 0 to 29, why did you censor the data after a score of 16 by merging all scores > 17? Please also indicate the probability of adverse outcomes for each score > 17.

5. It is of importance to provide in the manuscript a table and/or a figure summarizing the key findings of the first part of the study (derivation cohort).

6. Please further discuss your score in the light of the existing literature, not only by considering its ability to predict adverse outcomes (c-statistic) but also by considering its potential added-value (relevance of variables, ease of use…). In addition, the ability of your score to predict adverse outcomes should be also further discussed especially the sensitivity and specificity you found.

7. It would have been very interesting to compare your score to the results of the PAINTED study and to the scores developed for influenza pandemic. Please discuss this specific point.

6. PLOS authors have the option to publish the peer review history of their article (what does this mean?). If published, this will include your full peer review and any attached files.

Reviewer #1: **Yes: **Jonathan Ball

Reviewer #2: No

---

## [Author Response · Author response to Decision Letter 0]

22 Dec 2020

Thank you for considering our paper and providing the reviewer’s thoughtful comments. We have addressed them in a revised version of our paper. Our responses to each specific comment are as follows:

Editor:

I do wish to emphasize one point, the authors need to emphasize their uniqueness among various other scores recently published.

RESPONSE: We have extensively rewritten the discussion to address this point. The second paragraph has been added to describe the methodological features, including the study population selection criteria and choice of outcome, that ensure we have developed an optimal score. The third paragraph describes how our score improves upon previously recommended scores, specifically NEWS2 and PMEWS. We have rewritten paragraph five, critiquing other prediction scores, and we have added to paragraph six on rapid clinical scores to highlight their limitations compared to ours.

Reviewer #1:

1. The patient data was derived during the first wave of COVID-19 in the UK. The second wave in the UK, and elsewhere, appears to have some different characteristics. Could the authors comment on this and / or speculate on the need to repeat their data collection in other countries / during the current (second) wave, to establish whether it c statistic is affected?

RESPONSE: We have added a sentence to the last paragraph of the discussion suggesting this as a future research priority.

2. Might the authors data have a significant selection bias in that the location of data collection was emergency departments? Patients will have presented to primary care, Acute Medical Units etc etc hence how certain can we be that this result is generalisable?

RESPONSE: We agree that our data are most applicable to patients presenting to the ED and less applicable to patients presenting elsewhere. We have added consideration of this to the last paragraph of the discussion, along with a research recommendation.

3. Do the authors have any data to suggest what the short-term outcome of their patients was - e.g. how many were receiving level 1, level 2 and level 3 care at 72 hours post presentation? If not, might this secondary outcome be useful in future studies?

RESPONSE: We have added initial location of admission (ICU, HDU or ward) to Table 1.

4. Could the authors please include the data for patient sex in Table 2?

RESPONSE: We have added patient sex to Table 2.

5. Could the authors comment on the fact that ~67% of patients were admitted but only 31% were SARS-CoV-2 positive?

RESPONSE: We have added a sentence to the methods clarify that only admitted patients were tested at the participating sites during the first wave and noting in the results that the 31% positivity rate reflects a combination of lack of testing in those discharged, suboptimal sensitivity of standard tests, and the difficulty of differentiating COVID-19 from similar presentations. We feel this represents an expected rate for a clinically relevant cohort with suspected rather than confirmed COVID-19 (see response to reviewer #2, Q1).

6. Could the authors comment further on the very high proportion of missing data related to supplemental oxygen, and the much larger proportion in the validation cohort?

RESPONSE: We recognised that missing data could be a problem, given that oxygen supplementation is often poorly recorded in clinical notes, and planned multiple analyses using different approaches to handling missing data. We identify this as a limitation in the discussion – “Recording of inspired oxygen concentration was subject to a particularly high rate of missing data. We anticipated this problem and pre-specified analyses involving multiple imputation, deterministic imputation, and complete case analysis to explore the impact of missing data. There was reasonable concordance between the models”. We randomly allocated sites to the derivation and validation cohorts, so the higher rate of missing data in the validation cohort reflects random allocation of sites with higher missing data to the validation cohort.

7. Could the authors please provide more detail in figure 3, in particular, the number of patients with each score and the probabilities for each score 17-29?

RESPONSE: We have added the number of patients with each score and the probabilities for each score 17-29.

8. Given that the authors cut off for risk of a "bad outcome" is stated as a score >4 / probability of >9%, which can be achieved by being a 50 year old male with a temperature of 38.1 and a heart rate of 91 seems poorly calibrated to real life?

RESPONSE: We have added to paragraph eight of the discussion to address this important point. The key issue is that the score was developed upon, and therefore applies to, people who have attended the ED with suspected COVID-19. A patient with the characteristics described who did not need to attend the ED would probably not have a high risk of adverse outcome. However, our findings show that in patients attending the ED with a high baseline risks even people with a little or no physiological abnormality have a significant risk of adverse outcome. This does not necessarily mean they have to be admitted, but they should not be discharged with false reassurance that they are at low risk.

9. As currently written, I am unconvinced that the authors addition of age, sex and performance status adds anything useful to NEWS2 as a triage tool, especially in the absence of knowing whether the patient is positive for SARS-CoV-2. Perhaps they could clarify their justification for adding these variables to their real-world, pre-diagnosis target population?

RESPONSE: We have added the third paragraph of the discussion to address this point. Adding these variables improves the prediction, as measured by the c-statistic. The practical impact of this is to ensure that pre-morbid risk factors are taken into account when using NEWS2. Thus, a young person with unrestricted performance status is identified as being at low risk of adverse outcome, even if their NEWS2 score is 3-4, whereas an older person is identified as being at higher risk of adverse outcome, even if their NEWS2 score is 0-2. Decision-making on the basis of NEWS2 alone is likely to result in over-triage of younger, healthy people to unnecessary admission, and under-triage of older people with restricted performance status to inappropriate discharge.

Reviewer #2: 

The study is well-written and easy to read. Methods are well described and explained and the statistical analysis is appropriate. Results are interesting and the score could be useful in clinical practice in the event of a third pandemic wave. The two main strengths of the study are the large sample size and the appropriate method for building the score. One of the main limitations is the added-value of this new score, compared to existing triage scores for patients admitted to emergency department. I have some additional concerns that need to be discussed.

RESPONSE: We have extensively rewritten the discussion, with the addition of paragraphs two and three, to describe the added value of the new score.

1. Why did you consider patients with suspected and not confirmed COVID-19? In fact, this score could be used for all patients admitted to the emergency department, regardless of their COVID-19 status and is therefore not very different from existing triage scores. It would have been more interesting to develop a specific score for COVID-19 patients. Please clarify this point. In this regard, it would be interesting to repeat the analysis only in patients with confirmed COVID-19 if you are convinced of the need to develop specific predictive scores for COVID-19 patients.

RESPONSE: We have added to the first paragraph of the introduction to explain why we selected those with suspected COVID-19 and added to the second paragraph of the discussion why we feel this is a strength of our study. ED triage tools need to be used prospectively, when the diagnosis of COVID-19 is still only suspected in most cases, rather than retrospectively, when it has been confirmed. Rapid tests may be used to confirm COVID-19 in the ED, but limited sensitivity means COVID-19 cannot be ruled out in cases with a strong clinical suspicion, so COVID-19 still needs to be suspected in the absence of positive testing. Furthermore, despite much talk of a roll-out of rapid tests, we are now well into the second wave of the pandemic and use of rapid tests remains limited.

It would be interesting to repeat the analysis only in those with confirmed COVID-19 but the lack of testing on those discharged would mean that this effectively limits the cohort to those admitted, thus incurring the issues associated with limiting the cohort to patients whose selection for admission is likely to be based on key predictor variables.

2. What was the proportion of patients in whom COVID-19 was confirmed? This is a very important point before discussing your results, since the rationale for your study is based on the need to develop scores for COVID-19. Your results cannot be considered in the same way according to the proportion of confirmed COVID-19 patients.

RESPONSE: Table 1 shows that COVID-19 was confirmed in 31.1% of the derivation cohort and 31.4% of the validation cohort. We have added a sentence to the methods to clarify that only admitted patients were tested and noting in the results that the 31% positivity rate reflects a combination of lack of testing in those discharged, suboptimal sensitivity of standard tests, and the difficulty of differentiating COVID-19 from similar presentations. We feel this is an expected rate for a clinically-relevant cohort with suspected rather than confirmed COVID-19.

3. You state in the opening of the discussion that you included the NEWS2 score in your own score in addition to age, sex and performance status. However, you never described the NEWS2 score in the manuscript and in the Figure 1 (which is a Table), which is very confusing for readers. From my point of view, it would be clearer to simply indicate all the variable you included in your score rather than talking about the News2 score. Please clearly explain what the NEWS2 score is and clarify this point.

RESPONSE: We have added Appendix 3 to describe the NEWS2 score.

4. In figure 2, since your score ranged from 0 to 29, why did you censor the data after a score of 16 by merging all scores > 17? Please also indicate the probability of adverse outcomes for each score > 17.

RESPONSE: We collapsed scores above 17 into one column because the numbers in each strata above 17 were relatively small, and thus subject to greater random variation. We have amended the figure to include separate scores above 17.

5. It is of importance to provide in the manuscript a table and/or a figure summarizing the key findings of the first part of the study (derivation cohort).

RESPONSE: We have added a new figure (Figure 1) summarising the derivation part of the study and have renumbered subsequent figures.

6. Please further discuss your score in the light of the existing literature, not only by considering its ability to predict adverse outcomes (c-statistic) but also by considering its potential added-value (relevance of variables, ease of use…). In addition, the ability of your score to predict adverse outcomes should be also further discussed especially the sensitivity and specificity you found.

RESPONSE: We have added paragraph three of the discussion and extended paragraph five to discuss our score in the light of existing literature and consider its potential added value, and have extended paragraph eight to discuss how it could be used to predict adverse outcome.

7. It would have been very interesting to compare your score to the results of the PAINTED study and to the scores developed for influenza pandemic. Please discuss this specific point.

RESPONSE: We have added paragraph three of the discussion to address this point. We used data from our previous analysis of existing scores in the PRIEST data to compare our score to those developed for the influenza pandemic, since this provides much greater precision than the PAINTED data and a direct comparison using the same cohort.

---

## [Decision Letter · Decision Letter 1]

4 Jan 2021

PONE-D-20-34978R1

Derivation and validation of a clinical severity score for acutely ill adults with suspected COVID-19: The PRIEST observational cohort study

PLOS ONE

Dear Dr. Goodacre,

Thank you for submitting your manuscript to PLOS ONE. Before making the last decision, I wish you would address the issues raised by one of the reviewer's whose comments are attached below.

We look forward to receiving your revised manuscript.

Kind regards,

Itamar Ashkenazi

Academic Editor

PLOS ONE

Reviewers' comments:

Reviewer's Responses to Questions

**Comments to the Author**

1. If the authors have adequately addressed your comments raised in a previous round of review and you feel that this manuscript is now acceptable for publication, you may indicate that here to bypass the “Comments to the Author” section, enter your conflict of interest statement in the “Confidential to Editor” section, and submit your "Accept" recommendation.

Reviewer #1: (No Response)

Reviewer #2: All comments have been addressed

2. Is the manuscript technically sound, and do the data support the conclusions?

Reviewer #1: No

Reviewer #2: Yes

3. Has the statistical analysis been performed appropriately and rigorously? 

Reviewer #1: Yes

Reviewer #2: Yes

4. Have the authors made all data underlying the findings in their manuscript fully available?

Reviewer #1: Yes

Reviewer #2: Yes

5. Is the manuscript presented in an intelligible fashion and written in standard English?

Reviewer #1: Yes

Reviewer #2: Yes

6. Review Comments to the Author

Reviewer #1: Thank you for considering the points raised by both reviewers and making related changes to the manuscript.

I have the following residual concerns.

1. The primary outcome has been chosen for pragmatic reasons however predicting death or the need for organ support in the 30 days following presentation to ED ignores at least 2 very important groups, namely those who were admitted but did not required organ support, and those that did not receive organ support but did die after 30 days.

2. A minor point related to the penultimate paragraph on page 6, please include the total number of deaths observed - all that is stated is that the number was >200 [being 1% of the 20,000 used in the undocumented power calculation].

3. The proportion of missing data and the large difference between the derivation and validation cohorts in the "air or supplemental oxygen" question are problematic.

4. The first sentence of the first paragraph in the discussion is missing several vital caveats as follows:

"We have developed a clinical illness severity score for acutely ill patients, WHO PRESENTED TO EMERGENCY DEPARTMENTS IN THE UK BETWEEN MARCH AND JUNE[??] with suspected COVID-19" the clinical context, geography and calendar are critical.

" . . . to predict the risk of death or receipt of organ support IN THE FOLLOWING 30 DAYS"

". . . predicting adverse outcome with A high sensitivity BUT A VERY LOW SPECIFICITY."

5. The authors appear to be be simultaneously proposing their score be used to inform triage decisions in ED with a clear bias towards overtriage i.e. admitting some patients who at very low risk of the adverse outcome they define; and yet give a clinical example in which "patient preference" rather than clinical judgement should be a relevant factor. The latter assumes that the triage decision is not being made in the clinical context of a stressed / overwhelmed hospital system.

6. Similarly, the authors make the case that their score is superior to certain others because it includes acute physiology plus age and performance status but seem to believe that clinicians do not take these chronic factors into account when using acute scores to assist in objectively grading severity of illness. Furthermore, do the authors really believe that their c-statistic of 0.80 is clinically rather than statistically better than the 0.75 and 0.77 for CURB-65 and NEWS2?

7. The exclusion of any discriminating laboratory / diagnostic tests from a population who have attended a hospital appears to be an effort to provide a means of turning patients away at first triage. The authors have not demonstrated that the addition of any such variables fails to improve the discrimination, especially the specificity of their score. Given that accurate and rapid point-of-care testing is now widely available, is turnaround time in ED too critical to have considered the potential value of such tests?

8. The authors should comment on the QCOVID score - https://pubmed.ncbi.nlm.nih.gov/33082154/ - and consider the accompanying editorial.

Reviewer #2: Dear authors, you have taken into account all my comments and suggestions and provided a detailed point-by-point response. The manuscript has been significantly improved and I have no additional comments. Best regards.

7. PLOS authors have the option to publish the peer review history of their article (what does this mean?). If published, this will include your full peer review and any attached files.

Reviewer #1: **Yes: **Jonathan Ball

Reviewer #2: No

---

## [Author Response · Author response to Decision Letter 1]

4 Jan 2021

Thank you for considering our paper and providing the additional reviewer comments. We have revised the paper to address them and outline our responses to each point below.

Reviewer #1: Thank you for considering the points raised by both reviewers and making related changes to the manuscript.

I have the following residual concerns.

1. The primary outcome has been chosen for pragmatic reasons however predicting death or the need for organ support in the 30 days following presentation to ED ignores at least 2 very important groups, namely those who were admitted but did not required organ support, and those that did not receive organ support but did die after 30 days.

RESPONSE: We have added this to the limitations section of the discussion.

2. A minor point related to the penultimate paragraph on page 6, please include the total number of deaths observed - all that is stated is that the number was >200 [being 1% of the 20,000 used in the undocumented power calculation].

RESPONSE: We have added the adverse outcome rate in our cohort to this paragraph (not the death rate, since the sample size estimate was based on 200 adverse events, rather than deaths).

3. The proportion of missing data and the large difference between the derivation and validation cohorts in the "air or supplemental oxygen" question are problematic.

RESPONSE: We acknowledge in the discussion that the high rate of missing data for "air or supplemental oxygen" is an important limitation and describe the approaches we took to handling missing data. We randomly allocated sites to derivation and validation cohorts, so the difference between the cohorts is unlikely to be systematic.

4. The first sentence of the first paragraph in the discussion is missing several vital caveats as follows:

"We have developed a clinical illness severity score for acutely ill patients, WHO PRESENTED TO EMERGENCY DEPARTMENTS IN THE UK BETWEEN MARCH AND JUNE[??] with suspected COVID-19" the clinical context, geography and calendar are critical.

" . . . to predict the risk of death or receipt of organ support IN THE FOLLOWING 30 DAYS"

". . . predicting adverse outcome with A high sensitivity BUT A VERY LOW SPECIFICITY."

RESPONSE: We have added these caveats, with the exception of the location and timing of data collection. Science inevitably involves generalizing findings beyond the analysed data. In the discussion we consider potential limitations on generalizability and the need to validate in other settings and waves of the pandemic.

5. The authors appear to be simultaneously proposing their score be used to inform triage decisions in ED with a clear bias towards overtriage i.e. admitting some patients who at very low risk of the adverse outcome they define; and yet give a clinical example in which "patient preference" rather than clinical judgement should be a relevant factor. The latter assumes that the triage decision is not being made in the clinical context of a stressed / overwhelmed hospital system.

RESPONSE: We address this issue in the discussion and have added availability of resources to the factors that need to be considered alongside risk of adverse outcome in clinical decision-making.

6. Similarly, the authors make the case that their score is superior to certain others because it includes acute physiology plus age and performance status but seem to believe that clinicians do not take these chronic factors into account when using acute scores to assist in objectively grading severity of illness. Furthermore, do the authors really believe that their c-statistic of 0.80 is clinically rather than statistically better than the 0.75 and 0.77 for CURB-65 and NEWS2?

RESPONSE: We believe that it is up to clinicians how they use our score, or any alternative, to predict adverse outcome and support decision-making. Our analysis presents estimates of prognostic accuracy to inform clinicians in making their choices. We believe that a c-statistic of 0.80 is clinically significantly better than a c-statistic of 0.75 or 0.77, but our belief is not important. What is important is that we have presented our findings in a transparent manner that allows readers to judge for themselves whether the additional complexity of our score compared to NEWS2 or CURB-65 is worth the improved prognostic value.

7. The exclusion of any discriminating laboratory / diagnostic tests from a population who have attended a hospital appears to be an effort to provide a means of turning patients away at first triage. The authors have not demonstrated that the addition of any such variables fails to improve the discrimination, especially the specificity of their score. Given that accurate and rapid point-of-care testing is now widely available, is turnaround time in ED too critical to have considered the potential value of such tests?

RESPONSE: We have described the reasons why we think that a clinical score (without laboratory or radiological tests) is most useful and of greatest interest to clinicians working in the emergency department. This is why we have made this the principal output of our analysis. We have data available to explore whether laboratory and radiological information can improve prediction, but for the reasons outlined in our paper, we have not made this a priority in our plans for analysis and dissemination.

8. The authors should comment on the QCOVID score - https://pubmed.ncbi.nlm.nih.gov/33082154/ - and consider the accompanying editorial.

RESPONSE: The QCOVID score was developed in a population cohort study to predict the risk of death from COVID-19 in the general population. It is clearly very useful for guiding public health interventions, such as shielding for high-risk individuals and targeting vaccination, but has a very different purpose to our score, which is aimed at predicting the risk of adverse outcome among people who are acutely ill with COVID-19.

Reviewer #2: Dear authors, you have taken into account all my comments and suggestions and provided a detailed point-by-point response. The manuscript has been significantly improved and I have no additional comments. Best regards.

RESPONSE: Thank you

---

## [Editor Report · Decision Letter 2]

11 Jan 2021

Derivation and validation of a clinical severity score for acutely ill adults with suspected COVID-19: The PRIEST observational cohort study

PONE-D-20-34978R2

Dear Dr. Goodacre,

We’re pleased to inform you that your manuscript has been judged scientifically suitable for publication and will be formally accepted for publication once it meets all outstanding technical requirements.

Kind regards,

Itamar Ashkenazi

Academic Editor

PLOS ONE
---

## [Editor Report · Acceptance letter]

15 Jan 2021

PONE-D-20-34978R2 

Derivation and validation of a clinical severity score for acutely ill adults with suspected COVID-19: The PRIEST observational cohort study 

Dear Dr. Goodacre:

I'm pleased to inform you that your manuscript has been deemed suitable for publication in PLOS ONE. Congratulations! Your manuscript is now with our production department. 

Kind regards, 

on behalf of

Dr. Itamar Ashkenazi 

Academic Editor

PLOS ONE